# Rh(III)-Catalyzed Annulation of Boc-Protected Benzamides with Diazo Compounds: Approach to Isocoumarins

**DOI:** 10.3390/molecules24050937

**Published:** 2019-03-07

**Authors:** Guangyu Dong, Chunpu Li, Hong Liu

**Affiliations:** 1State Key Laboratory of Drug Research, Shanghai Institute of Materia Medica, Chinese Academy of Sciences, 555 Zuchongzhi Road, Shanghai 201203, China; 18115120903@163.com (G.D.); lcp1681993@163.com (C.L.); 2University of Chinese Academy of Sciences, No.19A Yuquan Road, Beijing 100049, China; 3School of Life Science and Technology, ShanghaiTech University, 100 Haike Road, Shanghai 201210,China

**Keywords:** rhodium, annulation, Boc-protected benzamides

## Abstract

A mild rhodium-catalyzed annulation of Boc-protected benzamides with diazo compounds via C−C/C−O bond formation has been explored. In the presence of [Cp*RhCl_2_]_2_, AgSbF_6_ and Cs_2_CO_3_, Boc-protected benzamides can be effectively annulated to yield isocoumarins in 0.5–2 h.

## 1. Introduction

Isocoumarins are valuable structural subunits because of their wide presence in numerous natural and synthetic compounds that exhibit potent biological activities (Figure 1) [1,2,3,4]. Some approaches to synthesize isocoumarins have been developed. Of the reported methods, the transition metal mediated coupling reaction facilitated by preactived C-X or C-M reagents has been recognized as a way to furnish the isocoumarin ring [5,6,7,8,9]. Additionally, oxidative annulations of the carboxylic acid with alkynes could also be viewed as an alternate strategy [10,11,12,13,14,15,16]. No doubt these methods are useful and practical, but the application of these reactions is somehow limited due to its requirements of a stoichiometric amount of oxidants and unkind temperatures. Consequently, it is highly desirable to develop more efficient methodologies for the synthesis of isocoumarins.

Rhodium-catalyzed C-H activation/cyclization has recently been pursued for constructing diverse heterocyclic systems [17,18,19,20]. Within rhodium catalysis, amide has attracted attention because of their particularly stable functionality. In this context, diazo compounds, as versatile partners of amides for C-H activation or cyclization, are widely applied in the organic process [21,22,23,24,25,26,27]. Prior contributions in this area include Rh(III)-catalyzed cyclization of benzamides and diazo compounds to construct isoindolinones via C-C/C-N bond formation reported by groups of Rovis, Yu and Cramer [28,29,30]. Besides, Rh(III) or Ir(III) catalysts were found to be effective for the synthsis of isoquinolinones, which is followed by the C-H coupling of *N*-methoxybenzamides with diazo compounds [31,32]. Despite significant progress, most of the reported studies are limited to bear *N*-heterocycles along with the formation of a C-C/C-N bond. In sharp contrast, few examples of Rh(III)-catalyzed C-C/C-O bond formation related to C-H cyclization of aromatics with diazo compounds are developed. Notably, C-C/C-O bond formation via Rh(III)-catalyzed C-H activation/cyclization of aromatics with diazo compounds has been achieved in 2015(Scheme 1, eqn (1)) [33]. In 2016, Rh(III)-catalyzed C-H annulation of *N*-tosylacrylamides and diazo compounds via C-C or C-N cleavages has been reported (Scheme 1, eqn (2)) [34]. Recently, A simple Rh(III)-catalyzed C-H activation that uses cyclic 2-diazo-1,3-diketones as starting materials has been developed (Scheme 1, eqn (3)) [35]. Though these methods are of high synthetic value, restricted substrate scope and high temperature or long reaction time may preclude their widespread application. Thus, it is necessary to develop more methods to construct a C-C/C-O bond by C-H annulation. Given that the directing group (DG) is pivotal in the construction of various heterocycles, we thus focus on finding DGs which could offer a tool to form diverse molecules in the means of breaking the C-N bond. In response to this unmet need, we show herein that *N*-*tert*-butoxycarbonyl (N-Boc) benzamides together with diazo compounds, allows delivery of corresponding isocoumarins under Rh(III) catalysis via C-C/C-O bond formation (Scheme 1, (4)). In this protocol, the Boc-protected benzamides could serve as good substrates and the reaction proceeds with concomitant removal of NHBoc auxiliary to afford the corresponding isocoumarins with high efficacy in short time (0.5–2 h). In this work, the *tert*-butyl formylcarbamate group formally served as an oxidizing DG using the C-N bond as an internal oxidant. Meanwhile, *tert*-butyl formylcarbamate group is an important structural motif of many biologically active compounds. Our work may provide a method to enable the late-stage diversification of functional molecules with *tert*-butyl formylcarbamate groups.

## 2. Results and Discussion

### 2.1. Optimization of Reaction Conditions for Synthesis of Ethyl 3-Methyl-1-oxo-1H-Isochromene-4-Carboxylate ***3aa***

We initiated our investigation by evaluating the feasibility of the combination of *N*-Boc benzamides **1a** or other *N*-substituted benzamides **1a_1_**–**1a_5_** and ethyl diazoacetoacetate **2a**. As shown in Table 1, amide derivatives **1a_1_**–**1a_5_** failed to undergo the annulation (Table 1, entries 1–5), while *N*-Boc benzamides could be utilized. Therefore *N*-Boc benzamides **1a** was opted to be used for optimization with ethyl diazoacetoacetate **2a**. We hypothesize that the electron-withdrawing capability of the Boc group makes the C-N bond easy to break. Several additives were further screened (Table 1, entries 7–10). When Ag_2_O and AgOAc were added to the reaction, no evidence of **3aa** was observed, while **3aa** was obtained in 33% yield with AgSbF_6_. We then turned our attention to screen solvents and acetonitrile was found to give the higher yield (47%; Table 1, entry 11). However, the explored efficiency was deficient, so the kind of bases was further explored (Table 1, entries 13–15). To our delight, we found that Cs_2_CO_3_ accompanied by AgSbF_6_ could obtain the highest yield (80%; Table 1, entry 14). Based on this model, we then explored catalyst species Co, Ru, Ir (Table 1, entries 16–18). [Cp*RhCl_2_]_2_ was chosen to be employed as catalysts for this annulation according to the obtained yields listed on Table 1. Interestingly, product **3aa** was obtained in a slightly lower yield when the reaction was carried out under N_2_ atmosphere (Table 1, entry 19). Next, an attempt to lower the reaction temperature to 25 °C resulted in a decreased yield (Table 1, entry 20). Notably, there was no obvious change in yield, whether the reaction dealt with higher temperature or longer reaction time (Table 1, entries 21 and 23). Hence the optimized conditions of Rh(III)-Catalyzed annulation of Boc-protected benzamides with diazo compounds were as follows: 5 mmol% [Cp*RhCl_2_]_2_ as catalyst, 2.0 equiv Cs_2_CO_3_ and 15 mmol% AgSbF_6_ as the additives. The reaction performed best at 60 °C for 0.5 h under air with acetonitrile as the solvent.

### 2.2. Substrate Scope for the Boc-Protected Benzamides

With the optimized conditions in hand, we embarked on the investigation of the substrate scope of Boc-protected benzamides to test the generality of this C-H activation/annulation reaction (Table 2). It was found that a variety of amides could successfully cyclize to give the desired products in moderate to good yields. Electron-donating groups such as methyl, *tert*-butyl and methoxy group at the *para*-position of the benzene ring gave **3ba**, **3ea** and **3fa** in 87%, 79% and 81% yields, respectively. In contrast, electron-withdrawing groups such as CF_3_ and nitro group afforded **3ka** and **3la** in 22% and trace yields. Halogen substituents (F, Cl, Br and I) provided **3ga**–**3ja** in 48–72% yields, indicating that the products of this reaction were compatible in transition-metal-catalyzed coupling reactions. In particular, we were pleased to obtain the single crystal X-ray of **3ga** (see Appendix A, CCDC: 1891024). Presumably owing to the steric effect, *meta*-phenzyl-substituted benzamide provided **3pa** in 60% yield, while *para-* and *ortho-*phenzyl-substituted benzamides provided **3ba** and **3ra** in 87% and 65% yields. Notably, 1-naphthamide gave the desired product **3ua** in 51% yield. In regard to the heteroaromatic amides such as thiophene, delivered the desired products **3va** in 68% yields.

### 2.3. Substrate Scope for the Diazo Compounds

After the examination of benzamides, diazo coupling partners were also investigated, and the results are described in Table 3. When the R^2^ with phenyl groups were used, the **3ab**−**3ae** were given in 51−79% yields. R_2_ with a cyclopropyl and a cyclohexyl group gave **3af** and **3ag** in moderate yield. When R_2_ with branched alkyl groups such as an isopropyl and an ethyl group were used, the **3ai** and **3aj** were given in 45% and 61% yields. When R_3_ with a methyl, a *tert*-butyl, an *n*-propyl or a benzyl group was used, the **3ak**–**3ao** were given in 25–70% yields.

### 2.4. Gram-Scale Preparation and Derivatization of the Annulation Product

In addition, 60% yield was obtained for the gram-scale synthesis of **3aa**, thus offering a practical access to highly functionalized isocoumarins (Scheme 2a). The applications of the isocoumarins have been demonstrated in several derivatization reactions. Formation of the desired **4** was achieved by reaction with ammonium acetate. In order to be more similar to the structures of Figure 1, removal of carboxylic esters at 4-position has been carried out to give **5** in 75% yield.

### 2.5. Mechanism

To obtain a more mechanistic insight, further experiments were carried out (Scheme 3). When *p-*methoxy-Boc-benzamide **1f** was run in competition with the *p-*trifluoromethyl-Boc-benzamide **1k**, the reaction favors the electron-donating substituent of the benzamide, suggesting that electrophilic-type benzamide were inherently less reactive (Scheme 3a). To gain further insights, the kinetic-isotope effect (KIE) was studied in parallel and competition experiments (Scheme 3b,c). Experiments with the same amounts of **1a** and deuterium-labeled benzamide **1a**–***d*_5_** were conducted, and a k_H_/k_D_ value of 1.44 was obtained (Scheme 3b). Furthermore, separate reactions of **1a** or **1a**–***d*_5_** together with **2a** gave the corresponding products **3aa** and **3aa**–***d*_4_**, respectively, displaying the similar KIE value of 1.24 (Scheme 3c). These results suggested that C−H cleavage is likely involved in the rate-limiting step. Moreover, when **1a** was treated with diethyl 2-diazomalonate, corresponding products was not observed under the standard reaction conditions, indicating that the final lactonization of ketone is significant in the transformation (Scheme 3d).

Based on the preliminary mechanistic experiments and literature precedents, a plausible mechanistic pathway is proposed in Scheme 4. Firstly, the catalytically active Rhodium species **A** is formed from [Cp*RhCl_2_]_2_, then the C-H metalation takes place to afford five-membered cyclometalated Rhodium species **B** via a Rh(III)-catalyzed C(sp^2^)-H bond cleavage. After insertion of the diazo compounds, Rh(III)-carbene species **C** can be formed with extrusion of N_2_. Subsequently, the species **C** undergoes a migratory insertion of carbene into the Rh-C bond, provided six-membered species **D**. Then, complex D could undergo protonation to give intermediate **E**, which can tautomerize to intermediate **F**. Finally, lactonization of intermediate **F** could afford the desired isocoumarin product **3** and the active catalyst **A**.

## 3. Materials and Methods

### 3.1. Chemistry

**Reagents and Solvents**: Rh catalysts and additives were commercially available. PE refers to petroleum ether (b.p. 60–90 °C), EA refers to ethyl acetate. Acetonitrile was used to the HPLC grade. All commercially available reagents and reactants were used without purification unless otherwise noted.

**Chromatography**: Flash column chromatography was carried out using commercially available 200–300 mesh under pressure unless otherwise indicated. Gradient flash chromatography was conducted eluting with PE/EA, they are listed as volume/volume ratios.

**Data Collection**: Nuclear magnetic resonance (NMR) spectra were run on 500 MHz instrument. Chemical shifts were reported in parts per million (ppm, δ) downfield from tetramethylsilane. Data are reported as follows: chemical shift in ppm (δ), multiplicity (s = singlet, d = doublet, t = triplet, q = quartet, m = multiplet), coupling constant (Hz) and integration. Low- and high-resolution mass spectra (LRMS and HRMS) were measured on spectrometer.

### 3.2. Experimental Part Method

#### 3.2.1. General Procedure for the Synthesis of Isocoumarins and α-Pyrones

To a mixture of [Cp*RhCl_2_]_2_ (3 mg, 0.0049 mmol, 5 mol%) and AgSbF_6_ (5 mg, 0.015 mmol, 15 mol%) in acetonitrile (2 mL) was added *tert*-butyl benzoylcarbamate **1** (0.1 mmol), diazo compounds **2** (0.12 mmol), Cs_2_CO_3_(65 mg, 0.2 mmol, 2 equiv). The reaction mixture was stirred at 60 °C for 0.5–2 h and the progress was monitored using TLC detection. After completion of present reaction, the solvent was evaporated under reduced pressure and the residue passed through flash column chromatography on silica gel to afford the desired products **3**.

#### 3.2.2. Procedure for the Synthesis of Diazo Substrates

Diazo substrates were synthesized from the corresponding ketonic esters or 1,3 di-ketones as shown in Appendix A. **2a**–**2o** was synthesized according to the literatures [36].

To a solution of ketonic ester or 1,3-di-ketone (5 mmol) in CH_3_CN, 6 mmol TsN_3_ was added. Then the reaction mixture was cooled to 0 °C and a solution of DBU (6 mmol) in 10 mL CH_3_CN was added dropwise. Next, the reaction temperature was raised to room temperature. After stirring for 3 h, the residue was extracted with EA for 3 times. The combined organic layers were washed with water and brine sequentially, dried over Na_2_SO_4_, filtered and concentrated. The crude product was purified by flash chromatography on silica gel (PE: EA = 100:1) to afford the corresponding product in 50–90% yields.

#### 3.2.3. Procedure for the Synthesis of Benzoylcarbamate Derivatives [37]

To a solution of benzamide (4.1 mmol) in dichloromethane (10.0 mL) was slowly added oxalyl chloride (630 mg, 0.43 mL 4.94 mmol) at 0 °C. The reaction mixture was warmed to 50 °C and stirred for 1 h. After cooling to 0 °C, a solution of the corresponding alcohol in dichloromethane was added, which was stirred at that temperature for 2 h. The reaction mixture was quenched by the addition of sat. aq. NaHCO_3_ and then extracted with dichloromethane. The combined organic layer was washed with brine and dried over Na_2_SO_4_. The volatiles were evaporated and the resulting crude product was purified by silica gel chromatography (eluent: dichloromethane to dichloromethane/ethyl acetate = 9/1) to give white powder. **1a**, **1b** and **1f** were synthesized according to the literatures [37,38,39].

*tert-Butyl (4-ethylbenzoyl)carbamate* (**1c**). White solid. ^1^H NMR (500 MHz, DMSO-*d_6_*) δ 10.54 (s, 1H), 7.78 (d, *J* = 8.2 Hz, 2H), 7.31 (d, *J* = 8.1 Hz, 2H), 2.66 (q, *J* = 7.6 Hz, 2H), 1.47 (s, 9H), 1.19 (t, *J* = 7.6 Hz, 3H). ^13^C NMR (126 MHz, DMSO-*d_6_*) δ 165.7, 150.4, 148.6, 130.9, 128.4, 127.6, 80.6, 28.0, 27.7, 15.1. HRMS (ESI) calcd for [C_14_H_19_NO_3_+Na] 272.1365, found 272.1253.

*tert*-Butyl (4-isopropylbenzoyl)carbamate (**1d**). White solid. ^1^H NMR (500 MHz, DMSO-*d_6_*) δ 10.54 (s, 1H), 7.79 (d, *J* = 8.3 Hz, 2H), 7.34 (d, *J* = 8.3 Hz, 2H), 2.95 (dt, *J* = 13.8, 6.9 Hz, 1H), 1.47 (s, 9H), 1.22 (s, 3H), 1.21 (s, 3H). ^13^C NMR (126 MHz, DMSO) δ 165.7, 153.1, 150.4, 131.1, 128.4, 126.2, 80.6, 33.3, 27.7, 23.5. HRMS (ESI) calcd for [C_15_H_21_NO_3_+Na] 286.1521, found 286.1417.

*tert*-Butyl (4-(tert-butyl)benzoyl)carbamate (**1e**). White solid. ^1^H NMR (600 MHz, DMSO-*d_6_*) δ 10.57 (s, 1H), 7.79 (d, *J* = 7.6 Hz, 2H), 7.49 (d, *J* = 8.4 Hz, 2H), 1.47 (s, 9H), 1.30 (s, 9H). ^13^C NMR (126 MHz, DMSO-*d_6_*) δ 165.7, 155.3, 150.4, 130.7, 128.2, 125.1, 80.6, 34.7, 30.8, 27.7. HRMS (ESI) calcd for [C_18_H_23_NO_3_-H]^−^ 276.1678, found 276.1603.

*tert*-Butyl (4-fluorobenzoyl)carbamate (**1h**). White solid. ^1^H NMR (500 MHz, DMSO-*d_6_*) δ 10.67 (s, 1H), 7.92 (dd, *J* = 8.9, 5.5 Hz, 2H), 7.31 (t, *J* = 8.8 Hz, 2H), 1.48 (s, 9H). ^13^C NMR (126 MHz, DMSO-*d_6_*) δ 165.9, 165.4, 163.9, 150.8, 131.6 (d, *J* = 9.3 Hz), 130.5 (d, *J* = 2.6 Hz), 115.8, 115.6, 81.2, 28.2. HRMS (ESI) calcd for [C_12_H_14_FNO_3_+Na] 262.0958, found 262.0848.

*tert*-Butyl (4-chlorobenzoyl)carbamate (**1g**). White solid. ^1^H NMR (500 MHz, CD_3_OD) δ 10.76 (s, 1H), 7.77 (d, *J* = 8.5 Hz, 2H), 7.68 (d, *J* = 8.5 Hz, 2H), 1.47 (s, 8H). ^13^C NMR (126 MHz, DMSO-*d_6_*) δ 165.4, 150.61, 137.5, 132.6, 130.5, 128.6, 81.1, 28.0. HRMS (ESI) calcd for [C_12_H_14_ClNO_3_+Na] 278.0662, found 278.0561.

*tert*-Butyl (4-bromobenzoyl)carbamate (**1i**). White solid. ^1^H NMR (400 MHz, CD_3_OD) δ 10.76 (s, 1H), 7.78 (d, *J* = 8.5 Hz, 2H), 7.69 (d, *J* = 8.5 Hz, 2H), 1.47 (s, 9H). ^13^C NMR (126 MHz, DMSO-*d_6_*) δ 165.2, 150.2, 132.7, 131.2, 130.3, 126.1, 80.8, 27.7. HRMS (ESI) calcd for [C_12_H_14_BrNO_3_+Na] 278.0662, found 278.0561.

*tert-*Butyl (4-iodobenzoyl)carbamate (**1j**). White solid. ^1^H NMR (500 MHz, DMSO-*d_6_*) δ 10.72 (s, 1H), 7.89–7.84 (m, 2H), 7.64–7.59 (m, 2H), 1.47 (s, 9H). ^13^C NMR (126 MHz, DMSO-*d_6_*) δ 165.5, 150.3, 137.1, 132.9, 130.1, 100.4, 80.8, 27.7. HRMS (ESI) calcd for [C_12_H_14_INO_3_+Na] 370.0018, found 369.9917.

*tert-Butyl (4-(trifluoromethyl)benzoyl)carbamate* (**1k**). White solid. ^1^H NMR (500 MHz, DMSO-*d_6_*) δ 10.93 (s, 1H), 8.03 (d, *J* = 8.1 Hz, 2H), 7.87 (d, *J* = 8.2 Hz, 2H), 1.50 (s, 9H). ^13^C NMR (126 MHz, DMSO-*d_6_*) δ 165.3, 150.2, 137.5, 129.1, 125.2 (d, *J* = 3.6 Hz), 122.7, 81.0, 27.7. HRMS (ESI) calcd for [C_13_H_14_F_3_NO_3_+Na] 312.0926, found 312.0814.

*tert-Butyl (4-nitrobenzoyl)carbamate* (**1l**). White solid. ^1^H NMR (500 MHz, DMSO-*d_6_*) δ 11.04 (s, 1H), 8.32 (d, *J* = 8.8 Hz, 2H), 8.06 (d, *J* = 8.8 Hz, 2H), 1.50 (s, 9H). ^13^C NMR (126 MHz, DMSO-*d_6_*) δ 165.5, 150.5, 149.8, 139.7, 130.1, 123.7, 81.6, 28.1. HRMS (ESI) calcd for [C_12_H_11_N_2_O_5_+Na] 289.0903, found 289.0801.

*tert-Butyl (4-(benzyloxy)benzoyl)carbamate* (**1m**). White solid. ^1^H NMR (500 MHz, DMSO-*d_6_*) δ 10.47 (s, 1H), 7.85 (d, *J* = 8.7 Hz, 2H), 7.46 (d, *J* = 7.4 Hz, 2H), 7.40 (t, *J* = 7.4 Hz, 2H), 7.34 (t, *J* = 7.2 Hz, 1H), 7.08 (d, *J* = 8.7 Hz, 2H), 5.19 (s, 2H), 1.47 (s, 9H). ^13^C NMR (126 MHz, DMSO-*d_6_*) δ 165.1, 161.6, 150.5, 136.5, 130.40, 128.4, 127.9, 127.7, 125.7, 114.3, 80.5, 69.4, 27.7. HRMS (ESI) calcd for [C_19_H_21_NO_4_+Na] 350.1471, found 350.1353.

*tert-Butyl (4-(dimethylamino)benzoyl)carbamate* (**1n**). White solid. ^1^H NMR (500 MHz, DMSO-*d_6_*) δ 11.66 (s, 1H), 7.88 (d, *J* = 9.0 Hz, 2H), 6.74 (d, *J* = 9.1 Hz, 2H), 3.03 (s, 6H), 1.51 (s, 9H). ^13^C NMR (126 MHz, DMSO-*d_6_*) δ 165.5, 164.1, 160.4, 153.6, 130.6, 128.9, 116.1, 110.7, 83.3, 27.5. HRMS (ESI) calcd for [C_14_H_20_N_2_O_3_+Na] 287.1474, found 287.1353.

*tert-Butyl [1,1′-biphenyl]-4-carbonylcarbamate* (**1o**). White solid. ^1^H NMR (500 MHz, DMSO-*d_6_*) δ 10.70 (s, 1H), 7.95 (d, *J* = 8.2 Hz, 2H), 7.76 (dd, *J* = 25.2, 7.9 Hz, 4H), 7.50 (t, *J* = 7.6 Hz, 2H), 7.42 (t, *J* = 7.3 Hz, 1H), 1.49 (s, 9H). ^13^C NMR (126 MHz, DMSO-*d_6_*) δ 166.2, 150.9, 144.3, 139.4, 132.8, 129.5, 128.8, 127.4, 126.9, 81.2, 28.3. HRMS (ESI) calcd for [C_18_H_19_NO_3_+Na] 320.1365, found 320.1262.

*tert-Butyl (3-methylbenzoyl)carbamate* (**1p**). White solid. ^1^H NMR (500 MHz, DMSO-*d_6_*) δ 10.61 (s, 1H), 7.71–7.60 (m, 2H), 7.37 (dt, *J* = 15.0, 7.4 Hz, 2H), 2.36 (s, 3H), 1.47 (s, 9H). ^13^C NMR (126 MHz, DMSO-*d_6_*) δ 166.1, 150.4, 137.6, 133.5, 132.9, 128.7, 128.2, 125.4, 80.7, 27.8, 20.80. HRMS (ESI) calcd for [C_13_H_17_NO_3_-H]^−^ 234.1208, found 234.1132.

*tert-Butyl (3-chlorobenzoyl)carbamate* (**1q**). White solid. ^1^H NMR (500 MHz, DMSO-*d_6_*) δ 10.76 (s, 1H), 7.89 (s, 1H), 7.79 (d, *J* = 7.8 Hz, 1H), 7.65 (d, *J* = 8.0 Hz, 1H), 7.51 (t, *J* = 7.9 Hz, 1H), 1.47 (s, 9H). ^13^C NMR (126 MHz, DMSO-*d_6_*) δ 164.74, 150.16, 135.54, 133.04, 132.06, 130.24, 127.96, 126.94, 80.92, 27.71. HRMS (ESI) calcd for [C_12_H_14_ClNO_3_-H]^−^ 254.0662, found 254.0587.

*tert-Butyl (2-methylbenzoyl)carbamate* (**1r**). White solid. ^1^H NMR (500 MHz, DMSO-*d_6_*) δ 10.66 (s, 1H), 7.38–7.30 (m, 2H), 7.27–7.19 (m, 2H), 2.30 (s, 3H), 1.39 (s, 9H). ^13^C NMR (126 MHz, DMSO-*d_6_*) δ 169.6, 150.8, 136.4, 135.5, 130.8, 130.4, 127.6, 125.9, 81.16, 28.1, 19.6. HRMS (ESI) calcd for [C_13_H_17_NO_3_-H]^−^ 234.1208, found 234.1131.

*tert-Butyl (2-chlorobenzoyl)carbamate* (**1s**). White solid. ^1^H NMR (500 MHz, DMSO-*d_6_*) δ 10.92 (s, 1H), 7.50–7.42 (m, 3H), 7.41–7.36 (m, 1H), 1.35 (s, 9H). ^13^C NMR (126 MHz, DMSO-*d_6_*) δ 167.1, 150.1, 136.3, 131.02, 129.2, 128.4, 127.1, 81.2, 27.5. HRMS (ESI) calcd for [C_12_H_14_ClNO_3_-H]^−^ 254.0662, found 254.0583.

*tert-Butyl (3,4-dimethylbenzoyl)carbamate* (**1t**). White solid. ^1^H NMR (500 MHz, CDCl_3_) δ 7.93 (s, 1H), 7.59 (d, *J* = 1.1 Hz, 1H), 7.51 (dd, *J* = 7.9, 1.8 Hz, 1H), 7.20 (d, *J* = 7.8 Hz, 1H), 2.30 (s, 6H), 1.53 (s, 9H). ^13^C NMR (126 MHz, CDCl_3_) δ 165.3, 149.8, 142.4, 137.5, 130.9, 130.1, 128.9, 124.9, 82.7, 28.2, 20.1, 19.9. HRMS (ESI) calcd for [C_14_H_19_NO_3_-H]^−^ 248.1365, found 248.1287.

*tert-Butyl 2-naphthoylcarbamate* (**1u**). White solid. ^1^H NMR (500 MHz, DMSO-*d_6_*) δ 10.83 (s, 1H), 8.55 (s, 1H), 8.09 (d, *J* = 8.0 Hz, 1H), 8.02 (d, *J* = 8.5 Hz, 2H), 7.91 (dd, *J* = 8.6, 1.8 Hz, 1H), 7.73–7.60 (m, 2H), 1.53 (s, 9H). ^13^C NMR (126 MHz, DMSO-*d_6_*) δ 165.8, 150.1, 134.3, 131.5, 130.4, 128.8, 127.9, 127.6, 127.3, 126.5, 124.2, 80.4. HRMS (ESI) calcd for [C_16_H_17_NO_3_ + Na] 294.1208, found 294.1104.

*tert-Butyl thiophene-2-carbonylcarbamate* (**1v**). White solid. ^1^H NMR (500 MHz, DMSO-*d_6_*) δ 10.72 (s, 1H), 8.06 (d, *J* = 3.5 Hz, 1H), 7.92 (d, *J* = 4.7 Hz, 1H), 7.23–7.09 (m, 1H), 1.48 (s, 9H). ^13^C NMR (126 MHz, DMSO-*d_6_*) δ 159.8, 150.0, 138.6, 133.8, 131.0, 128.3, 80.9, 27.8. HRMS (ESI) calcd for [C_10_H_13_NO_3_S-H]^−^ 226.0616, found 226.0539.

**1a_1_**–**1a_5_** were synthesized according to the literatures [40,41,42,43].

#### 3.2.4. Characterization of the Products

*Ethyl 3-methyl-1-oxo-1H-isochromene-4-carboxylate* (**3aa**). Colorless oil. ^1^H NMR (500 MHz, DMSO-*d_6_*) δ 8.17 (dd, *J* = 7.9, 0.8 Hz, 1H), 7.93–7.81 (m, 1H), 7.72 (d, *J* = 8.1 Hz, 1H), 7.62 (t, *J* = 7.6 Hz, 1H), 4.41 (q, *J* = 7.1 Hz, 2H), 2.39 (s, 3H), 1.35 (t, *J* = 7.1 Hz, 3H). ^13^C NMR (126 MHz, CDCl_3_) δ 165.9, 161.3, 157.8, 135.2, 134.8, 129.8, 128.3, 124.2, 119.7, 110.4, 61.8, 19.4, 14.3. HRMS (ESI) calcd for [C_13_H_12_O_4_ + H]^+^ 233.0736, found 233.0808.

*Ethyl 3,6-dimethyl-1-oxo-1H-isochromene-4-carboxylate* (**3ba**). White solid. ^1^H NMR (500 MHz, CDCl_3_) δ 8.13 (d, *J* = 8.1 Hz, 1H), 7.50 (s, 1H), 7.28 (d, *J* = 8.1 Hz, 1H), 4.43 (q, *J* = 7.1 Hz, 2H), 2.45 (s, 3H), 2.41 (s, 3H), 1.41 (t, *J* = 7.2 Hz, 3H). ^13^C NMR (126 MHz, CDCl_3_) δ 165.9, 161.3, 157.5, 146.3, 134.7, 129.7, 129.5, 124.1, 117.1, 110.2, 61.7, 22.3, 19.3, 14.3. HRMS (ESI) calcd for [C_14_H_14_O_4_ + H]^+^ 247.0892, found 247.0960.

*Ethyl 6-ethyl-3-methyl-1-oxo-1H-isochromene-4-carboxylate* (**3ca**). White solid. ^1^H NMR (500 MHz, CDCl_3_) δ 8.18 (d, *J* = 8.1 Hz, 1H), 7.54 (s, 1H), 7.34 (d, *J* = 8.1 Hz, 1H), 4.45 (q, *J* = 7.1 Hz, 2H), 2.75 (q, *J* = 7.6 Hz, 2H), 2.43 (s, 3H), 1.42 (t, *J* = 7.2 Hz, 3H), 1.27 (t, *J* = 7.6 Hz, 3H). ^13^C NMR (126 MHz, CDCl_3_) δ 166.1, 161.4, 157.7, 152.5, 134.9, 129.9, 128.4, 123.1, 117.4, 110.4, 61.7, 29.6, 19.4, 15.1, 14.4. HRMS (ESI) calcd for [C_15_H_16_O_4_ + H]^+^ 261.1049, found 261.1125.

*Ethyl 6-isopropyl-3-methyl-1-oxo-1H-isochromene-4-carboxylate* (**3da**). White solid. ^1^H NMR (500 MHz, CDCl_3_) δ 8.20 (d, *J* = 8.2 Hz, 1H), 7.57 (d, *J* = 1.5 Hz, 1H), 7.38 (d, *J* = 8.2 Hz, 1H), 4.45 (q, *J* = 7.1 Hz, 2H), 3.01 (dt, *J* = 13.8, 6.9 Hz, 1H), 2.44 (s, 3H), 1.43 (t, *J* = 7.1 Hz, 3H), 1.29 (s, 3H), 1.27 (s, 3H). ^13^C NMR (126 MHz, CDCl_3_) δ 166.12, 161.37, 157.74, 156.99, 134.93, 129.96, 127.08, 121.84, 117.55, 110.48, 34.91, 23.66, 19.39, 14.39. HRMS (ESI) calcd for [C_16_H_18_O_4_ + H]^+^ 275.1205, found 275.1284.

*Ethyl 6-(tert-butyl)-3-methyl-1-oxo-1H-isochromene-4-carboxylate* (**3ea**). White solid. ^1^H NMR (500 MHz, CDCl_3_) δ 8.19 (d, *J* = 8.4 Hz, 1H), 7.75 (d, *J* = 1.7 Hz, 1H), 7.54 (dd, *J* = 8.4, 1.7 Hz, 1H), 4.45 (q, *J* = 7.1 Hz, 2H), 2.44 (s, 3H), 1.43 (t, *J* = 7.1 Hz, 3H), 1.35 (s, 9H). ^13^C NMR (126 MHz, CDCl_3_) δ 166.2, 161.3, 159.2, 157.8, 134.6, 129.5, 126.1, 120.6, 117.1, 110.6, 61.6, 35.7, 31.1, 19.4, 14.4. HRMS (ESI) calcd for [C_17_H_20_O_4_ + H]^+^ 289.1362, found 289.1437.

*Ethyl 6-methoxy-3-methyl-1-oxo-1H-isochromene-4-carboxylate* (**3fa**). White solid. ^1^H NMR (500 MHz, D_2_O) δ 8.18 (d, *J* = 8.8 Hz, 1H), 7.22 (d, *J* = 2.4 Hz, 1H), 7.02 (dd, *J* = 8.8, 2.4 Hz, 1H), 4.43 (q, *J* = 7.2 Hz, 2H), 3.89 (s, 3H), 2.44 (s, 3H), 1.42 (t, *J* = 7.2 Hz, 3H). ^13^C NMR (126 MHz, D_2_O) δ 166.0, 165.0, 161.0, 158.9, 137.0, 132.0, 116.3, 112.6, 110.0, 107.0, 61.7, 55.73, 19.7, 14.4. HRMS (ESI) calcd for [C_14_H_14_O_5_ + H]^+^ 263.0841, found 263.0911.

*Ethyl 6-chloro-3-methyl-1-oxo-1H-isochromene-4-carboxylate* (**3ga**). White solid. ^1^H NMR (500 MHz, CDCl_3_) δ 8.22 (dd, *J* = 8.5, 0.4 Hz, 1H), 7.86 (d, *J* = 1.4 Hz, 1H), 7.48 (dd, *J* = 8.5, 1.2 Hz, 1H), 4.48 (q, *J* = 7.1 Hz, 2H), 2.50 (s, 3H), 1.45 (t, *J* = 7.2 Hz, 3H). ^13^C NMR (126 MHz, CDCl_3_) δ 164.6, 159.7, 159.0, 141.4, 135.4, 130.6, 128.0, 123.5, 117.1, 108.6, 61.2, 19.0, 13.6. HRMS (ESI) calcd for [C_13_H_11_ClO_4_ + H]^+^ 267.0346, found 267.0415.

*Ethyl 6-fluoro-3-methyl-1-oxo-1H-isochromene-4-carboxylate* (**3ha**). White solid. ^1^H NMR (500 MHz, CDCl_3_) δ 8.29 (dd, *J* = 8.4, 6.2 Hz, 1H), 7.54 (d, *J* = 10.5 Hz, 1H), 7.23–7.13 (m, 1H), 4.44 (q, *J* = 7.1 Hz, 2H), 2.48 (s, 3H), 1.43 (t, *J* = 7.1 Hz, 3H). ^13^C NMR (126 MHz, CDCl_3_) δ 168.0, 166.2, 165.5, 160.3, 160.1, 137.5, 133.1, 116.4, 116.0, 110.8, 109.6 (d, *J* = 2.6 Hz), 61.9, 19.8, 14.4. HRMS (ESI) calcd for [C_13_H_11_FNO_3_ - H]^−^ 249.0641, found 249.0562.

*Ethyl 6-bromo-3-methyl-1-oxo-1H-isochromene-4-carboxylate* (**3ia**). White solid. ^1^H NMR (500 MHz, CDCl_3_) δ 8.12 (d, *J* = 8.5 Hz, 1H), 8.01 (d, *J* = 1.7 Hz, 1H), 7.62 (dd, *J* = 8.5, 1.8 Hz, 1H), 4.46 (q, *J* = 7.1 Hz, 2H), 2.48 (s, 3H), 1.43 (t, *J* = 7.2 Hz, 3H). ^13^C NMR (126 MHz, CDCl_3_) δ 165.1, 160.3, 159.5, 135.9, 131.5, 131.0, 130.8, 127.1, 118.0, 109.0, 61.7, 19.5, 14.1. HRMS (ESI) calcd for [C_13_H_12_BrO_4_ + H]^+^ 310.9841, found 310.9917.

*Ethyl 6-iodo-3-methyl-1-oxo-1H-isochromene-4-carboxylate* (**3ja**). White solid. ^1^H NMR (500 MHz, CDCl_3_) δ 8.20 (s, 1H), 7.93 (dd, *J* = 8.3, 2.0 Hz, 1H), 7.82 (d, *J* = 8.3 Hz, 1H), 4.48–4.41 (m, 2H), 2.46 (s, 3H), 1.43 (dd, *J* = 7.4, 6.9 Hz, 3H). ^13^C NMR (126 MHz, CDCl_3_) δ 165.3, 160.8, 159.5, 137.4, 135.8, 133.4, 130.8, 118.7, 109.0, 104.1, 62.0, 19.7, 14.3. HRMS (ESI) calcd for [C_13_H_12_IO_4_ + H]^+^ 358.9702, found 358.9773.

*Ethyl 3-methyl-1-oxo-6-(trifluoromethyl)-1H-isochromene-4-carboxylate* (**3ka**). White solid. ^1^H NMR (500 MHz, CDCl_3_) δ 8.40 (d, *J* = 8.3 Hz, 1H), 8.17 (s, 1H), 7.73 (d, *J* = 8.2 Hz, 1H), 4.48 (q, *J* = 7.1 Hz, 2H), 2.52 (s, 3H), 1.44 (t, *J* = 7.2 Hz, 3H). ^13^C NMR (126 MHz, CDCl_3_) δ 165.2, 160.3, 160.1, 136.7, 136.4, 135.3, 130.7, 124.5 (d, *J* = 3.3 Hz), 122.2–121.4 (m), 109.7, 62.1, 19.8, 14.3. HRMS (ESI) calcd for [C_14_H_11_F_3_O_4_ + H]^+^ 301.0609, found 301.0676.

*Ethyl 3-methyl-6-nitro-1-oxo-1H-isochromene-4-carboxylate* (**3la**). White solid. HRMS (ESI) calcd for [C_13_H_11_NO_6_ - H]^−^ 276.0586, found 276.0518.

*Ethyl 6-(benzyloxy)-3-methyl-1-oxo-1H-isochromene-4-carboxylate* (**3ma**). White solid. ^1^H NMR (500 MHz, CDCl_3_) δ 8.22 (d, *J* = 8.8 Hz, 1H), 7.49–7.39 (m, 4H), 7.36 (dd, *J* = 16.5, 4.7 Hz, 2H), 7.12 (dd, *J* = 8.8, 2.4 Hz, 1H), 5.18 (s, 2H), 4.45 (q, *J* = 7.1 Hz, 2H), 2.47 (s, 3H), 1.43 (t, *J* = 7.1 Hz, 3H). ^13^C NMR (126 MHz, CDCl_3_) δ 166.2, 164.4, 161.2, 159.1, 137.3, 136.1, 132.3, 129.1, 128.7, 127.9, 117.1, 113.1, 110.3, 108.3, 70.7, 61.9, 19.9, 14.6. HRMS (ESI) calcd for [C_20_H_18_IO_5_ + H]^+^ 339.1154, found 339.1230.

*Ethyl 6-(dimethylamino)-3-methyl-1-oxo-1H-isochromene-4-carboxylate* (**3na**). White solid. ^1^H NMR (500 MHz, CDCl_3_) δ 8.06 (d, *J* = 8.7 Hz, 1H), 6.78 (d, *J* = 9.9 Hz, 2H), 4.42 (q, *J* = 6.8 Hz, 2H), 3.08 (s, 6H), 2.39 (s, 3H), 1.41 (t, *J* = 7.1 Hz, 3H). ^13^C NMR (126 MHz, CDCl_3_) δ 166.5, 161.6, 158.1, 154.5, 136.3, 131.5, 112.7, 110.2, 107.5, 103.9, 61.4, 40.2, 19.5, 14.4. HRMS (ESI) calcd for [C_15_H_17_NO_4_ + H]^+^ 276.1158, found 276.1230.

*Ethyl 3-methyl-1-oxo-6-phenyl-1H-isochromene-4-carboxylate* (**3oa**). White solid. ^1^H NMR (500 MHz, CDCl_3_) δ 8.37–8.31 (m, 1H), 7.99 (d, *J* = 1.5 Hz, 1H), 7.73 (dd, *J* = 8.2, 1.7 Hz, 1H), 7.64 (dd, *J* = 7.9, 0.9 Hz, 2H), 7.49 (t, *J* = 7.4 Hz, 2H), 7.43 (t, *J* = 7.3 Hz, 1H), 4.47 (q, *J* = 7.1 Hz, 2H), 2.48 (s, 3H), 1.44 (t, *J* = 7.1 Hz, 3H). ^13^C NMR (126 MHz, CDCl_3_) δ 166.0, 161.3, 158.3, 148.0, 139.7, 135.2, 130.4, 129.2, 128.8, 127.6, 127.3, 122.6, 118.3, 110.4, 61.8, 19.5, 14.4. HRMS (ESI) calcd for [C_19_H_16_O_4_ + H]^+^ 309.1049, found 309.1120.

*Ethyl 3,7-dimethyl-1-oxo-1H-isochromene-4-carboxylate* (**3pa**). White solid. ^1^H NMR (500 MHz, CDCl_3_) δ 8.06 (s, 1H), 7.66 (d, *J* = 8.3 Hz, 1H), 7.53 (dd, *J* = 8.3, 1.5 Hz, 1H), 4.43 (q, *J* = 7.2 Hz, 2H), 2.44 (s, 3H), 2.43 (s, 3H), 1.41 (t, *J* = 7.2 Hz, 3H). ^13^C NMR (126 MHz, CDCl_3_) δ 166.0, 161.5, 157.0, 138.5, 136.4, 132.2, 129.4, 124.1, 119.5, 110.2, 61.7, 21.2, 19.3, 14.3. HRMS (ESI) calcd for [C_14_H_14_O_4_ + H]^+^ 247.0892, found 247.0961.

*Ethyl 7-chloro-3-methyl-1-oxo-1H-isochromene-4-carboxylate* (**3qa**). White solid. ^1^H NMR (500 MHz, CDCl_3_) δ 8.25 (s, 1H), 7.79 (d, *J* = 8.7 Hz, 1H), 7.68 (d, *J* = 8.7 Hz, 1H), 4.44 (q, *J* = 7.1 Hz, 2H), 2.47 (s, 3H), 1.42 (t, *J* = 7.1 Hz, 3H). ^13^C NMR (126 MHz, CDCl_3_) δ 165.4, 160.0, 158.5, 135.3, 134.1, 133.1, 129.0, 126.0, 120.8, 109.6, 61.8, 19.4, 14.2. HRMS (ESI) calcd for [C_13_H_11_ClO_4_ + H]^+^ 267.0346, found 267.0422.

*Ethyl 3,5-dimethyl-1-oxo-1H-isochromene-4-carboxylate* (**3ra**). White solid. ^1^H NMR (500 MHz, CDCl_3_) δ 7.59–7.53 (m, 1H), 7.49 (d, *J* = 7.8 Hz, 1H), 7.29 (d, *J* = 7.3 Hz, 1H), 4.43 (q, *J* = 7.1 Hz, 2H), 2.80 (s, 3H), 2.38 (s, 3H), 1.41 (t, *J* = 7.1 Hz, 3H). ^13^C NMR (126 MHz, CDCl_3_) δ 166.4, 160.6, 156.6, 143.81, 136.2, 134.3, 131.2, 121.9, 118.1, 110.9, 61.7, 23.6, 19.0, 14.3. HRMS (ESI) calcd for [C_14_H_14_O_4_ + H]^+^ 247.0892, found 247.0960.

*Ethyl 8-chloro-3-methyl-1-oxo-1H-isochromene-4-carboxylate* (**3sa**). White solid. ^1^H NMR (500 MHz, CDCl_3_) δ 7.63–7.56 (m, 2H), 7.53 (d, *J* = 7.6 Hz, 1H), 4.44 (q, *J* = 7.2 Hz, 2H), 2.41 (s, 3H), 1.42 (t, *J* = 7.2 Hz, 3H). ^13^C NMR (126 MHz, CDCl_3_) δ 165.8, 158.0, 157.7, 137.7, 137.3, 134.8, 131.2, 122.9, 116.8, 110.3, 62.0, 19.2, 14.3. HRMS (ESI) calcd for [C_13_H_11_ClO_4_ + H]^+^ 267.0346, found 267.0425.

*Ethyl 3,6,7-trimethyl-1-oxo-1H-isochromene-4-carboxylate* (**3ta**). White solid. ^1^H NMR (500 MHz, CDCl_3_) δ 8.02 (s, 1H), 7.51 (s, 1H), 4.44 (q, *J* = 7.1 Hz, 2H), 2.42 (s, 3H), 2.37 (s, 3H), 2.35 (s, 3H), 1.42 (t, *J* = 7.2 Hz, 3H). ^13^C NMR (126 MHz, CDCl_3_) δ 166.0, 161.4, 156.8, 145.4, 137.6, 132.6, 129.8, 124.6, 117.3, 110.0, 61.5, 20.7, 19.6, 19.2, 14.2. HRMS (ESI) calcd for [C_15_H_16_O_4_ + H]^+^ 261.1049, found 261.1116.

*Ethyl 3-methyl-1-oxo-1H-benzo[g]isochromene-4-carboxylate* (**3ua**). White solid. ^1^H NMR (500 MHz, CDCl_3_) δ 8.97–8.93 (m, 1H), 8.26–8.21 (m, 1H), 8.07–8.02 (m, 1H), 7.98–7.92 (m, 1H), 7.71–7.62 (m, 1H), 7.62–7.55 (m, 1H), 4.58–4.50 (m, 2H), 2.52–2.49 (m, 3H), 1.53–1.47 (m, 3H). ^13^C NMR (126 MHz, CDCl_3_) δ 166.3, 161.6, 156.1, 136.6, 132.2, 132.1, 129.8, 129.7, 129.6, 129.2, 128.5, 127.2, 123.1, 110.3, 61.8, 19.4, 14.4. HRMS (ESI) calcd for [C_17_H_14_O_4_ + H]^+^ 283.0892, found 283.0966.

*Ethyl 5-methyl-7-oxo-7H-thieno [2,3-c]pyran-4-carboxylate* (**3vq**). White solid. ^1^H NMR (500 MHz, CDCl_3_) δ 7.87 (d, *J* = 5.2 Hz, 1H), 7.75 (d, *J* = 5.2 Hz, 1H), 4.45 (q, *J* = 7.1 Hz, 2H), 2.68 (s, 3H), 1.45 (t, *J* = 7.1 Hz, 3H). ^13^C NMR (126 MHz, CDCl_3_) δ 165.0, 164.5, 157.2, 145.4, 136.7, 126.3, 122.4, 108.3, 61.7, 20.2, 14.4. HRMS (ESI) calcd for [C_11_H_10_O_4_S + H]^+^ 283.0300, found 239.0371.

*Ethyl 1-oxo-3-phenyl-1H-isochromene-4-carboxylate* (**3ab**). White solid. ^1^H NMR (500 MHz, CDCl_3_) δ 8.37 (dd, *J* = 8.0, 0.8 Hz, 1H), 7.82–7.77 (m, 1H), 7.75 (dd, *J* = 8.1, 0.7 Hz, 1H), 7.65 (dd, *J* = 8.0, 1.6 Hz, 2H), 7.61–7.55 (m, 1H), 7.50–7.43 (m, 3H), 4.20 (q, *J* = 7.2 Hz, 2H), 1.05 (t, *J* = 7.1 Hz, 3H). ^13^C NMR (126 MHz, CDCl_3_) δ 166.3, 161.1, 155.4, 135.3, 134.7, 132.6, 130.5, 129.9, 128.8, 128.5, 128.1, 124.1, 119.8, 110.9, 61.9, 13.5. HRMS (ESI) calcd for [C_18_H_14_O_4_ + H]^+^ 295.0892, found 295.0961.

*Ethyl 3-(4-methoxyphenyl)-1-oxo-1H-isochromene-4-carboxylate* (**3ac**). white solid.^1^H NMR (500 MHz, CDCl_3_) δ 8.34 (d, *J* = 7.9 Hz, 1H), 7.77 (t, *J* = 7.6 Hz, 1H), 7.70 (d, *J* = 8.1 Hz, 1H), 7.61 (d, *J* = 8.7 Hz, 2H), 7.55 (t, *J* = 7.6 Hz, 1H), 6.96 (d, *J* = 8.7 Hz, 2H), 4.24 (q, *J* = 7.1 Hz, 2H), 3.86 (s, 3H), 1.12 (t, *J* = 7.1 Hz, 3H). ^13^C NMR (126 MHz, CDCl_3_) δ 166.6, 161.4, 161.2, 155.1, 135.2, 134.9, 129.8, 128.4, 124.8, 123.9, 119.5, 113.9, 109.9, 61.9, 55.4, 13.7. HRMS (ESI) calcd for [C_19_H_16_O_5_ + H]^+^ 325.0998, found 325.1069.

*Ethyl 3-(4-chlorophenyl)-1-oxo-1H-isochromene-4-carboxylate* (**3ad**). White solid. ^1^H NMR (500 MHz, CDCl_3_) δ 8.36 (dd, *J* = 7.9, 0.9 Hz, 1H), 7.82–7.77 (m, 1H), 7.72 (d, *J* = 7.7 Hz, 1H), 7.62–7.56 (m, 3H), 7.47–7.41 (m, 2H), 4.23 (q, *J* = 7.2 Hz, 2H), 1.12 (t, *J* = 7.2 Hz, 3H). ^13^C NMR (126 MHz, CDCl_3_) δ 166.0, 160.8, 153.9, 136.7, 135.3, 134.4, 130.9, 129.9, 129.4, 129.0, 128.7, 124.1, 119.7, 111.2, 62.0, 13.6. HRMS (ESI) calcd for [C_18_H_13_ClO_4_ + H]^+^ 329.0502, found 329.0575.

*Ethyl 3-(naphthalen-2-yl)-1-oxo-1H-isochromene-4-carboxylate* (**3ae**). Colorless oil. ^1^H NMR (500 MHz, CDCl_3_) δ 8.39 (dd, *J* = 7.9, 0.7 Hz, 1H), 8.20 (s, 1H), 7.95–7.86 (m, 3H), 7.84–7.75 (m, 2H), 7.71 (dd, *J* = 8.6, 1.7 Hz, 1H), 7.63–7.51 (m, 3H), 4.21 (q, *J* = 7.1 Hz, 2H), 0.99 (t, *J* = 7.1 Hz, 3H). ^13^C NMR (126 MHz, CDCl_3_) δ 166.5, 161.2, 155.3, 135.4, 134.9, 134.1, 132.8, 129.9, 128.9, 128.7, 128.3, 127.8, 127.0, 124.8, 124.3, 119.9, 111.3, 62.1, 13.7. HRMS (ESI) calcd for [C_22_H_16_O_4_ + H]^+^ 345.1049, found 345.1111.

*Ethyl 3-cyclopropyl-1-oxo-1H-isochromene-4-carboxylate* (**3af**). Colorless oil. ^1^H NMR (500 MHz, CDCl_3_) δ 8.25 (ddd, *J* = 8.0, 1.4, 0.7 Hz, 2H), 7.76–7.65 (m, 4H), 7.47 (ddd, *J* = 8.4, 6.8, 1.6 Hz, 2H), 4.49 (q, *J* = 7.2 Hz, 4H), 2.31 (tt, *J* = 8.3, 5.0 Hz, 2H), 1.45 (t, *J* = 7.1 Hz, 6H), 1.33–1.22 (m, 5H), 1.09–0.97 (m, 4H). ^13^C NMR (126 MHz, CDCl_3_) δ 166.2, 160.8, 160.3, 135.1, 129.7, 127.6, 123.6, 119.1, 109.4, 61.7, 14.3, 12.5, 8.5. HRMS (ESI) calcd for [C_15_H_14_O_4_ + H]^+^ 259.0892, found 259.0957.

*Ethyl 3-cyclohexyl-1-oxo-1H-isochromene-4-carboxylate* (**3ag**). Colorless oil. ^1^H NMR (500 MHz, CDCl_3_) δ 8.34–8.28 (m, 1H), 7.77–7.72 (m, 1H), 7.63 (d, *J* = 8.1 Hz, 1H), 7.52 (dd, *J* = 11.2, 4.0 Hz, 1H), 4.47 (q, *J* = 7.1 Hz, 2H), 2.79 (tt, *J* = 11.8, 3.2 Hz, 1H), 1.87 (d, *J* = 10.2 Hz, 4H), 1.84–1.71 (m, 3H), 1.45 (t, *J* = 7.1 Hz, 3H), 1.32 (dd, *J* = 15.5, 5.9 Hz, 3H). ^13^C NMR (126 MHz, CDCl_3_) δ 166.4, 163.1, 161.8, 135.3, 135.1, 130.0, 128.4, 124.2, 120.0, 109.6, 62.1, 42.1, 30.3, 26.3, 25.8, 14.6. HRMS (ESI) calcd for [C_18_H_20_O_4_ + H]^+^ 301.1362, found 301.1433.

*7,8,9,10-tetrahydrocyclohepta[c]isochromene-5,11-dione* (**3ah**). White solid. ^1^H NMR (500 MHz, CD_3_OD) δ 8.20 (s, 1H), 8.07 (s, 1H), 7.76 (d, *J* = 6.4 Hz, 1H), 7.53 (d, *J* = 6.5 Hz, 1H), 2.96 (s, 2H), 2.81 (d, *J* = 3.1 Hz, 2H), 1.94 (d, *J* = 2.5 Hz, 4H). ^13^C NMR (126 MHz, CD_3_OD) δ 204.4, 165.5, 162.6, 136.2, 135.9, 130.2, 129.2, 125.7, 120.8, 117.3, 43.7, 33.1, 24.2, 23.4. HRMS (ESI) calcd for [C_14_H_12_O_3_ + H]^+^ 227.0786, found 227.0708.

*Ethyl 3-isopropyl-1-oxo-1H-isochromene-4-carboxylate* (**3ai**). Colorless oil. ^1^H NMR (500 MHz, CDCl_3_) δ 8.28 (dd, *J* = 7.9, 1.0 Hz, 1H), 7.74–7.69 (m, 1H), 7.59 (d, *J* = 8.1 Hz, 1H), 7.52–7.47 (m, 1H), 4.44 (q, *J* = 7.2 Hz, 2H), 3.13 (dt, *J* = 13.7, 6.8 Hz, 1H), 1.41 (t, *J* = 7.2 Hz, 3H), 1.32 (s, 3H), 1.31 (s, 3H). ^13^C NMR (126 MHz, CDCl_3_) δ 166.1, 163.2, 161.5, 135.1, 134.8, 129.8, 128.3, 124.0, 119.8, 109.2, 61.9, 31.7, 20.2, 14.3. HRMS (ESI) calcd for [C_15_H_16_O_4_ + H]^+^ 261.1049, found 261.1123.

*Ethyl 3-ethyl-1-oxo-1H-isochromene-4-carboxylate* (**3aj**). Colorless oil. ^1^H NMR (500 MHz, CDCl_3_) δ 8.29 (d, *J* = 7.8 Hz, 1H), 7.76–7.68 (m, 2H), 7.53–7.48 (m, 1H), 4.44 (q, *J* = 7.2 Hz, 2H), 2.71 (q, *J* = 7.5 Hz, 2H), 1.42 (t, *J* = 7.1 Hz, 3H), 1.33 (t, *J* = 7.5 Hz, 3H). ^13^C NMR (126 MHz, CDCl_3_) δ 165.9, 161.6, 135.1, 134.8, 129.8, 128.3, 124.2, 119.7, 109.8, 61.8, 26.4, 14.3, 12.3. HRMS (ESI) calcd for [C_14_H_14_O_4_ + H]^+^ 247.0892, found 247.0964.

*tert-Butyl 3-methyl-1-oxo-1H-isochromene-4-carboxylate* (**3ak**). White solid. ^1^H NMR (500 MHz, CDCl_3_) δ 8.30 (d, *J* = 8.0 Hz, 1H), 7.81–7.69 (m, 2H), 7.52 (t, *J* = 7.2 Hz, 1H), 2.45 (s, 3H), 1.65 (s, 9H). ^13^C NMR (126 MHz, CDCl_3_) δ 165.1, 161.5, 156.3, 135.1, 135.0, 129.8, 128.1, 123.9, 119.7, 111.8, 83.2, 28.3, 19.1. HRMS (ESI) calcd for [C_15_H_16_O_4_ + H]^+^ 261.1049, found 261.1114.

*Benzyl 3-methyl-1-oxo-1H-isochromene-4-carboxylate* (**3al**). White solid. ^1^H NMR (500 MHz, CDCl_3_) δ 8.27 (d, *J* = 8.0 Hz, 1H), 7.70 (ddd, *J* = 11.6, 9.6, 4.7 Hz, 2H), 7.52–7.43 (m, 3H), 7.43–7.34 (m, 3H), 5.42 (s, 2H), 2.40 (s, 3H). ^13^C NMR (126 MHz, CDCl_3_) δ 165.7, 161.1, 158.0, 135.1, 134.6, 129.7, 128.9, 128.6, 128.2, 124.1, 119.5, 109.99, 67.6, 19.4. HRMS (ESI) calcd for [C_18_H_14_O_4_ + H]^+^ 295.0892, found 295.0962.

*Allyl 3-methyl-1-oxo-1H-isochromene-4-carboxylate* (**3am**). Colorless oil. ^1^H NMR (500 MHz, CDCl_3_) δ 8.27 (dd, *J* = 27.2, 8.0 Hz, 1H), 7.80–7.70 (m, 2H), 7.51 (t, *J* = 7.5 Hz, 1H), 6.05 (ddt, *J* = 16.5, 10.4, 6.0 Hz, 1H), 5.45 (dd, *J* = 17.2, 1.3 Hz, 1H), 5.35 (dd, *J* = 10.4, 0.9 Hz, 1H), 4.88 (d, *J* = 6.0 Hz, 2H), 2.46 (s, 3H). ^13^C NMR (126 MHz, CDCl_3_) δ 165.6, 161.2, 158.1, 135.2, 134.6, 131.3, 129.7, 128.3, 124.2, 119.8, 119.5, 110.0, 66.4, 19.4. HRMS (ESI) calcd for [C_14_H_12_O_4_ + H]^+^ 245.0736, found 245.0811.

*Methyl 3-methyl-1-oxo-1H-isochromene-4-carboxylate* (**3an**). Colorless oil. ^1^H NMR (500 MHz, CDCl_3_) δ 8.31–8.25 (m, 1H), 7.73 (qd, *J* = 8.4, 4.0 Hz, 2H), 3.97 (s, 3H), 2.45 (s, 3H). ^13^C NMR (126 MHz, CDCl_3_) δ 166.3, 161.1, 158.1, 135.1, 134.5, 129.7, 128.2, 124.2, 119.5, 110.0, 52.4, 19.4. HRMS (ESI) calcd for [C_12_H_10_O_4_ + H]^+^ 219.0579, found 219.0648.

*Propyl 3-methyl-1-oxo-1H-isochromene-4-carboxylate* (**3ao**). Colorless oil. ^1^H NMR (500 MHz, CDCl_3_) δ 8.31–8.26 (m, 1H), 7.74 (qd, *J* = 8.3, 4.0 Hz, 2H), 7.51 (ddd, *J* = 8.2, 6.7, 1.8 Hz, 1H), 4.35 (t, *J* = 6.7 Hz, 2H), 2.46 (s, 3H), 1.87–[44]1.75 (m, 2H), 1.04 (t, *J* = 7.4 Hz, 3H).^13^C NMR (126 MHz, CDCl_3_) δ 166.1, 161.4, 157.8, 135.2, 134.8, 129.8, 128.3, 124.3, 119.7, 110.5, 67.5, 22.1, 19.5, 10.7. HRMS (ESI) calcd for [C_14_H_14_O_4_ + H]^+^ 247.0892, found 247.0964.

#### 3.2.5. Derivatization Experiments

*Ethyl 3-methyl-1-oxo-1,2-dihydroisoquinoline-4-carboxylate* (**4**) [45]. A mixture of the product **3aa** (323 mg) and ammonium acetate (250 mg) in 0.5 mL of acetic acid is stirred at 80 °C overnight, and then cooled and poured into water. The solid is collected by filtration, washed with water and dried to yield the title compound [44]. White solid. ^1^H NMR (400 MHz, CDCl_3_) δ 11.74 (s, 1H), 8.42 (dd, *J* = 8.0, 1.0 Hz, 1H), 7.94 (d, *J* = 8.3 Hz, 1H), 7.74–7.65 (m, 1H), 7.48 (t, *J* = 7.2 Hz, 1H), 4.46 (d, *J* = 7.1 Hz, 2H), 2.59 (s, 3H), 1.44 (t, *J* = 7.1 Hz, 3H). 13C NMR (126MHz, CDCl_3_) δ 167.2, 141.2, 135.9, 133.4, 127.4, 126.5, 124.5, 109.5, 61.4, 19.1, 14.5. HRMS (ESI) calcd for [C_13_H_13_NO_3_ + H]^+^ 232.0895, found 232.0970.

*3-methyl-1H-isochromen-1-one* (**5**) [46]. To a 100 mL rd-bottom flask was charged with conc. HCl (3.51 mL, 96 mmol) and heated to 110 °C. **3aa** (232 mg, 1 mmol) was then added portionwise and the mixture was allowed to stir at reflux for 22 h. The resulting white solid and yellow solution was cooled to room temperature and the solid was collected by vacuum filtration. The solid was further dried using a mechanical vacuum pump (10^−3^ torr), and then heated over a hot water bath at 50–60 °C for 1 h to give isocoumarin **5** as an off-white solid (120 mg, 75%) [47]. White solid. ^1^H NMR (400 MHz, CDCl_3_) δ 8.16 (d, *J* = 8.0 Hz, 1H), 7.61 (td, *J* = 7.7, 1.3 Hz, 1H), 7.42–7.33 (m, 1H), 7.27 (d, *J* = 8.0 Hz, 1H), 6.19 (s, 1H), 2.21 (d, *J* = 0.6 Hz, 3H). ^13^C NMR (126 MHz, CDCl_3_) δ 163.1, 154.7, 137.8, 134.8, 129.6, 127.6, 124.9, 120.0, 103.6, 19.7. HRMS (ESI) calcd for [C_10_H_8_O_2_ + H]^+^ 161.0524, found 161.0593.

#### 3.2.6. Compete Experiments and Mechanistic Studies

**1a**–**d_5_**. Benzoic acid-d_5_ (10 mmol) in SOCl_2_ (10 mL) was refluxed at 85 °C. After 30 min, the mixture turned clear. The reaction was stopped, and the reaction solution was concentrated to give the acyl chloride. A solution of acyl chloride prepared above in anhydrous dichloromethane (10 mL) was injected dropwise to an aqueous ammonia solution (con. 20 mL) in an ice bath. After stirring for 30 min, the precipitate was collected by suction filtration, washed with water and *n*-hexane, and dried under reduced pressure at 50 °C. Recrystallization from ethyl acetate afforded the compounds benzamide following general Procedure for the synthesis of benzoylcarbamate derivatives. ^1^H NMR (500 MHz, DMSO-*d_6_*) δ 10.65 (s, 1H), 1.47 (s, 10H). ^13^C NMR (126 MHz, DMSO-*d_6_*) δ 166.47, 150.87, 133.82, 132.31, 128.35, 81.17, 28.20. HRMS (ESI) calcd for [C_12_H_10_D_5_NO_3_ + Na]^+^ 249.1366, found 249.1256.

**Scheme 3b**: To a mixture of [Cp*RhCl_2_]_2_ (3 mg, 0.0049 mmol, 5 mol%) and AgSbF_6_ (5 mg, 0.015 mmol, 15 mol%) in acetonitrile (2 mL) was added **1a** (22.1 mg, 0.6 mmol), **1a**-***d*_5_**(22.6 mg, 0.1mmol) and ethyl 2-diazo-3-oxobutanoate **2a** (18.7 mg, 0.12 mmol), Cs_2_CO_3_(65 mg, 0.2 mmol, 2 equiv). The reaction mixture was stirred at 60 °C for 1 min and the progress was monitored using TLC detection. After completion of the present reaction, the solvent was evaporated under reduced pressure and the residue passed through flash column chromatography on silica gel to afford the mixture of products **3a** and **3a**–**d_4_** with 17.0 mg (74% yield).

**Scheme 3c**: To a mixture of [Cp*RhCl_2_]_2_ (3 mg, 0.0049 mmol, 5 mol%) and AgSbF_6_ (5 mg, 0.015 mmol, 15 mol%) in acetonitrile (2 mL) was added **1a** (22.1 mg, 0.6 mmol) or **1a**–**d_5_** (22.6 mg, 0.1mmol) and ethyl 2-diazo-3-oxobutanoate **2a** (18.7 mg, 0.12 mmol), Cs_2_CO_3_(65 mg, 0.2 mmol, 2 equiv). The reaction mixture was stirred at 60 °C for 1 min and the progress was monitored using TLC detection. After completion of the present reaction, the solvent was evaporated under reduced pressure and the residue passed through flash column chromatography on silica gel to afford the mixture of products **3aa** (17.0 mg, 74%) or **3aa**–**d_4_** (14.1 mg, 59.7%).

**Scheme 3d**: To a mixture of [Cp*RhCl_2_]_2_ (3 mg, 0.0049 mmol, 5 mol%) and AgSbF_6_ (5 mg, 0.015 mmol, 15 mol%) in acetonitrile (2 mL) was added **1a** (22 mg, 0.1 mmol) and diethyl 2-diazomalonate (22 mg, 0.12 mmol), Cs_2_CO_3_ (65 mg, 0.2 mmol, 2 equiv). The reaction mixture was stirred at 80 °C and the progress was monitored using TLC detection.

## 4. Conclusions

In conclusion, we have successfully developed a Rhodium-catalyzed C-H activation/annulation of diazo compounds with Boc-protected benzamidesubstrates for efficient synthesis of isocoumarins. The *tert*-butyl formylcarbamate group formally served as an oxidizing DG using the C-N bond as an internal oxidant. In this strategy, the novel Boc-amide groups as removable directing groups enable the benzamides to construct C-C/C-O bonds to provide isocoumarins. Moreover, this reaction features broad substrate scopes and good tolerance. We believe the mild procedure will be of importance to medicinal chemists.

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
