# Peer review of "Rh(III)-Catalyzed Annulation of Boc-Protected Benzamides with Diazo Compounds: Approach to Isocoumarins"

_molecules, 2019, doi:10.3390/molecules24050937_

Round 1
Reviewer 1 Report
This manuscript describes the new synthetic method of isocoumarins. The authors used BOC-protected benzamides and diazo compounds as a starting materials and a rhodium(III) complex as a catalyst. The authors state that this method can greatly shorten the reaction time. The synthesis and characterization of the compounds are almost fine and reasonable. I think this synthetic method is useful for organic synthesis, therefore, I feel the manuscript can be publishable to Molecules with minor revisions.
1. The compounds were characterized by 1H and 13C NMR and MS spectra. The MS spectra of the compounds should be included in the Supporting Information.
2.In Scheme 4, detail of E to A should be drawn.
Author Response
Reviewer 1:
This manuscript describes the new synthetic method of isocoumarins. The authors used BOC-protected benzamides and diazo compounds as a starting materials and a rhodium(III) complex as a catalyst. The authors state that this method can greatly shorten the reaction time. The synthesis and characterization of the compounds are almost fine and reasonable. I think this synthetic method is useful for organic synthesis, therefore, I feel the manuscript can be publishable to Molecules with minor revisions.
Question 1: The compounds were characterized by 1H and 13C NMR and MS spectra. The MS spectra of the compounds should be included in the Supporting Information.
Response: Thanks a lot for your suggestions. We have added MS spectra of the compounds into the Supporting Information. Taken 3c as example (in the attachment).
Question 2: In Scheme 4, detail of E to A should be drawn.
Response: Thanks a lot for your suggestions. We have drawn detail of E to A. The complex D could undergo protonation to give intermediate E, which can tautomerize to intermediate F. Finally, lactonization of intermediate F could afford the desired isocoumarin product 3 and the active catalyst A. I hope to meet your requirements. And we have made corrections in our revised manuscript.

Reviewer 2 Report
This manuscript describes that Rh(III)-catalyzed isocoumarin-formation of benzamides with alpha-diazo beta-ketoesters is highly accelerated by Boc-protection of the benzamides. The tendency would be clear and the characterization of compounds is fine. Therefore, the results in this manuscript can be published in Molecules as a publication after minor revision. The authors should consider following points.
1) The authors should give some explanation for the reason for acceleration effect arising from Boc-protection.
2) Page 2, Scheme 1, eq 1: The reference of this equation is ref. 35, however, which shows the reaction of N-aryl amidines instead of benzamides. The authors should check it again.
3) In all structural formulas, the numbers in “R1, R2, R3•••” should be typed in super script face in place of subscript face for being readily distinguished from numbers in the formula such as “CO2Me and N2”.
4) Line 62: Please delete one of two “the”s in front of “feasibility.”
5) Lines 62-63 and 65: “N-(tert-butyl)benzamides” should be “N-(tert-butyloxycarbonyl)benzamides” or “N-Boc benzamides.”
6) Page 3, Table 1: Substituents R1 and R2 should be specified for 1a1-1a5. Otherwise, the readers cannot understand which is which.
7) Lines 68-73: The explanation for entries 7-15 should be placed before that for entries 16-18 which is in lines 66-68.
8) Lines 128: “more” appears to be “less.” Otherwise, this sentence holds contradiction.
9) Lines 131-132: “Significant primary kinetic isotope effects were observed (kH/kD= 1.44), which suggested that the C−H bond cleavage was not likely to be involved in the rate-limiting step.” The former part of this sentence appears to be in contradiction with the later part.
10) Pages16-19: For all refs, abbreviation of journal names should be used and issue numbers should be removed. For example, “The Journal of organic chemistry 1999, 64, (24), 8770-8779.” should be “J. Org. Chem. 1999, 64, 8770-8779.”
Author Response
Reviewer 2:
This manuscript describes that Rh(III)-catalyzed isocoumarin-formation of benzamides with alpha-diazo beta-ketoesters is highly accelerated by Boc-protection of the benzamides. The tendency would be clear and the characterization of compounds is fine. Therefore, the results in this manuscript can be published in Molecules as a publication after minor revision. The authors should consider following points.
Question 1: The authors should give some explanation for the reason for acceleration effect arising from Boc-protection.
Response: Thanks a lot for your suggestions. Based on the plausible reaction mechanism, lactonization of intermediate F could afford the desired isocoumarin product 3. In this step, the electron-withdrawing capability of Boc group makes the C-N bond easy to break. This may be the reason for acceleration effect arising from Boc-protection. We have added this explanation in our revised manuscript.
Question 2: Page 2, Scheme 1, eq 1: The reference of this equation is ref. 35, however, which shows the reaction of N-aryl amidines instead of benzamides. The authors should check it again.
Response: Thanks a lot for your corrections. We have changed Page 2, Scheme 1, eq 3: the reference of this equation, ref. 35 into the right reference.
Question 3: In all structural formulas, the numbers in “R1, R2, R3•••” should be typed in super script face in place of subscript face for being readily distinguished from numbers in the formula such as “CO2Me and N2”.
Response: Thanks a lot for your corrections. We have changed “R1, R2, R3•••” to be typed in super script face.
Question 4: Line 62: Please delete one of two “the”s in front of “feasibility”.
Response: Thanks a lot for your corrections. We have deleted Line 62: one of two“the”s in front of“feasibility”.
Question 5: Lines 62-63 and 65: “N-(tert-butyl)benzamides” should be “N-(tert-butyloxycarbonyl)benzamides” or “N-Boc benzamides.”
Response: Thanks a lot for your corrections. We have changed Lines 62-63 and 65: “N-(tert-butyl)benzamides”to“N-Boc benzamides.”
Question 6: Page 3, Table 1: Substituents R1 and R2 should be specified for 1a1-1a5. Otherwise, the readers cannot understand which is which.
Response: Thanks a lot for your suggestions.We have specified Page 3, Table 1: Substituents R1 and R2 for 1a1-1a5.
Question 7: Lines 68-73: The explanation for entries 7-15 should be placed before that for entries 16-18 which is in lines 66-68.
Response: Thanks a lot for your corrections. We have placed Lines 68-73: the explanation for entries 7-15 before that for entries 16-18 which is in lines 66-68.
Question 8: Lines 128: the reaction favors the electron-donating substituent of the benzamide, suggesting that electrophilic-type benzamide were inherently more reactive (Scheme 3a). “more” appears to be “less.” Otherwise, this sentence holds contradiction.
Response: Thanks a lot for your suggestions. We have changed Lines 128: “more” to“less.”
Question 9: Lines 131-132: “Significant primary kinetic isotope effects were observed (kH/kD= 1.44), which suggested that the C−H bond cleavage was not likely to be involved in the rate-limiting step.” The former part of this sentence appears to be in contradiction with the later part.
Response: Thanks a lot for your suggestions. We have made changes in our revised manuscript as follows:
Experiments with the same amounts of 1a and deuterium-labeled benzamide 1a-d5 were conducted, and a kH/kD value of 1.44 was obtained (Scheme 3b). Furthermore separate reactions of 1a or 1a-d5 together with 2a gave the corresponding products 3aa and 3aa-d4, respectively, displaying the similar KIE value of 1.24 (Scheme 3c). These results suggested that C−H cleavage is likely involved in the rate-limiting step.
Question 10: Pages16-19: For all refs, abbreviation of journal names should be used and issue numbers should be removed. For example, “The Journal of organic chemistry 1999, 64, (24), 8770-8779.” should be “J. Org. Chem. 1999, 64, 8770-8779.”
Response: Thanks a lot for your corrections. We have used abbreviation of journal names for all refs and removed issue numbers.
Reviewer 3 Report
In this manuscript, Liu and coworkers presented a rhodium(III)-catalyzed annulation of benzamides with diazo compounds. Although similar transformations [rhodium(III)-catalyzed annulation of benzamides with diazo compounds via C(sp2)-H activation] had been extensively studied, the use of Boc-protected benzamides enabled efficient annulation within relatively short reaction time. There are a few points that I would like the authors to address prior to publication.
(1) The advance of using NHBoc as the directing & leaving group should be discussed in further details.
(2) All the annulation products bear substitutions at the 4-position of isocoumarins, but this can not be found in any of the molecules shown in Figure 1.
(3) Related to Comment (2), the authors need to showcase further transformations of the annulation products.
Author Response
Reviewer 3:
In this manuscript, Liu and coworkers presented a rhodium(III)-catalyzed annulation of benzamides with diazo compounds. Although similar transformations [rhodium(III)-catalyzed annulation of benzamides with diazo compounds via C(sp2)-H activation] had been extensively studied, the use of Boc-protected benzamides enabled efficient annulation within relatively short reaction time. There are a few points that I would like the authors to address prior to publication.
Question 1: The advance of using NHBoc as the directing & leaving group should be discussed in further details.
Response: Thanks a lot for your suggestions. Besides the Boc-protected benzamides are easily obtained under mild conditions, there are many advantages of this directing & leaving group. In our work, the tert-butyl formylcarbamate group formally served as an oxidizing DG using the C–N bond as an internal oxidant without any need for additives. Meanwhile, tert-butyl formylcarbamate group is an important structural motif of many biologically active compounds (For example, compound 56 in J. Med. Chem. 2014, 57, 5348−5355). Our work may provide a method to enable the late-stage diversifcation of functional molecules with tert-butyl formylcarbamate groups. We have added these discussions in our revised manuscript.
Question 2: All the annulation products bear substitutions at the 4-position of isocoumarins, but this can not be found in any of the molecules shown in Figure 1.
Response: Thanks a lot for your suggestions. We have found that most of the reported bioactive isocoumarin structures with no substitution at 4, but our 4-position of annulation products related to Comment (3) could be removed under certain conditions.
Question 3: Related to Comment (2), the authors need to showcase further transformations of the annulation products.
Response: Thanks a lot for your suggestions. We have showcase further transformations of the annulation products in Scheme 2. Formation of the desired 4 was achieved by reaction with ammonium acetate. In order to be more similar to the structures of Figure 1, removal of carboxylic esters at 4-position has been carried out to give 5 in 75% yield. We believe that will help in the search of new biologically active compounds and drug discovery.

Round 2
Reviewer 3 Report
The authors have revised the manuscript according to my suggestions. I would recommend it for publication in Molecules.